# Tomography of the London Underground:
# a Scalable Model for Origin-Destination Data

**Nicolò Colombo**
Department of Statistical Science
University College London
nicolo.colombo@ucl.ac.uk

**Ricardo Silva**
The Alan Turing Institute and
Department of Statistical Science
University College London
ricardo.silva@ucl.ac.uk

**Soong Kang**
School of Management
University College London
smkang@ucl.ac.uk

## Abstract

The paper addresses the classical network tomography problem of inferring local traffic given origin-destination observations. Focusing on large complex public transportation systems, we build a scalable model that exploits input-output information to estimate the unobserved link/station loads and the users' path preferences. Based on the reconstruction of the users' travel time distribution, the model is flexible enough to capture possible different path-choice strategies and correlations between users travelling on similar paths at similar times. The corresponding likelihood function is intractable for medium or large-scale networks and we propose two distinct strategies, namely the exact maximum-likelihood inference of an approximate but tractable model and the variational inference of the original intractable model. As an application of our approach, we consider the emblematic case of the London underground network, where a tap-in/tap-out system tracks the starting/exit time and location of all journeys in a day. A set of synthetic simulations and real data provided by Transport For London are used to validate and test the model on the predictions of observable and unobservable quantities.

## 1 Introduction

In the last decades, networks have been playing an increasingly important role in our all-day lives [1, 2, 3, 4, 5, 6]. Most of the time, networks cannot be inspected directly and their properties should be reconstructed form end-point or partial and local observations [7, 8]. The problem has been referred to as network 'tomography', a medical word to denote clinical techniques that produce detailed images of the interior of the body from external signals [9, 10]. Nowadays the concept of tomography has gained wider meanings and the idea applies, in different forms, to many kinds of communication and transportation networks [11, 12, 13]. In particular, as the availability of huge amounts of data has grown exponentially, network tomography has become an important branch of statistical modelling [14, 15, 16, 17, 8]. However, due to the complexity of the task, existing methods are usually only designed for small-size networks and become intractable for most real-world applications (see [7, 18] for a discussion on this point). The case of large public transportation networks has attracted special attention since massive datasets of input-output single-user data have been produced by tap-in and tap-out systems installed in big city as London, Singapore and Beijing [19, 20, 18, 21].

Depending on the available measurements, two complementary formulations of network tomography have been considered: (i) the reconstruction of origin-destination distributions from local and partial traffic observations [11, 14, 9, 15, 16] and (ii) the estimation of the link and node loads from input-output information [22, 23, 24]. In practice, the knowledge of the unobserved quantities may help design structural improvements of the network or be used to predict the system's behaviour in case of disruptions [25, 26, 13, 27, 28]. Focusing on the second (also referred to as 'dual') formulation of the tomography problem, this paper addresses the challenging case where both the amount of data and the size of the network are large. When only aggregated data are observable, traffic flows over a given network can also be analysed by methods such as collective graphical models for diffusion dynamics [29, 30].

An important real-world application of dual network tomography is reconstructing the traffic of bits sent from a source node to a destination node in a network of servers, terminals and routers. The usual assumption, in those cases, is the tree structure of the network and models infer the bits trajectories from a series of local delays, *i.e.* loss functions defined at each location in the network [22, 23, 24]. The posterior of the travel time distribution at each intermediate position along the path is then used to reconstruct the unobserved local loads, *i.e.* the number of packets at a given node and time. We extend and apply this general idea to urban public transportation systems. The traffic to be estimated is the flow of people travelling across the system during a day, *i.e.* the number of people at a given location and time (station/link load). The nodes of the network are ($> 100$) underground stations, connected via ($\sim 10$) partially overlapping underground 'lines', which can be look at as interacting 'layers' of connectivity [31]. The observations are single-user records with information about the origin, destination, starting time and exit time of each journey. Two key unobserved quantities to be estimated are (i) the users' path preferences for a given origin-destination pair [32, 28] and (ii) the station/link loads [33, 34, 35]. Put together, a model for the users' path preferences and a precise estimation of the local train loads can help detect network anomalies or predict the behaviour of the system in case of previously unobserved disruptions [18, 27, 21].

Respect to the classical communication network case, modelling a complex transportation system requires three challenging extensions: (i) the network structure is a multi-layer (loopy) network, where users are allowed to 'change line' on those nodes that are shared by different layers; (ii) the user's choice between many feasible paths follows rules that can go far beyond simple length-related schemes; (iii) harder physical constraints (the train time schedule for example) may create high correlations between users travelling on the same path at similar times. Taking into a account such peculiar features of transportation networks, while keeping the model scalable respect to both the size of the network and the dataset, is the main contribution of this work.

**Model outline**   We represent the transportation system by a sparse graph, where each node is associated with an underground station and each edge with a physical connection between two stations. The full network is the sum of simple sub-graphs (lines) connected by sets of shared nodes (where the users are allowed to change line) [31]. For a given origin-destination pair, there may exist a finite number of possible simple (non redundant) trajectories, corresponding to distinct line-change strategies. The unobserved user's choice is treated as a latent variable taking values over the set of all feasible paths between the origin and destination. The corresponding probability distribution may depend on the length of the path, *i.e.* the number of nodes crossed by the path, or any other arbitrary feature of the path. In our multi-layers setup, for example, it is natural to include a 'depth' parameter taking into account the number of layers visited, *i.e.* the number of lines changes.

For any feasible path $\gamma = [\gamma_1, \ldots, \gamma_\ell]$, the travel time at the intermediate stations is defined by the recursive relation

$$t(\gamma_i) \sim t(\gamma_{i-1}) + \text{Poisson}(a(\gamma_{i-1}, s_o + t(\gamma_{i-1}), \gamma)) \qquad i = 1, \ldots, \ell \qquad (1)$$

where $t(x)$ the is travel time at location $x \in \{\gamma_1, \ldots, \gamma_\ell\}$, $s_o$ is the starting time, $a = a(x, s_o + t(x), \gamma)$ are local delays that depend on the location, $x$, the absolute time $s_o + t(x)$ and the path $\gamma$. The choice of the Poisson distribution is convenient [1] in this framework due to its simple single-parameter form and the fact that $t(x)$ is an integer in the dataset that motivates this work (travel time is recorded in minutes). The dependence on $\gamma$ allows including global path-related features, such as, for example, an extra delay associated to each line change along the path or the time spent by the user while walking through the origin and destination stations. The dependence on $s_o$ and $t(x)$ is what ensures

the scalability of the model because all users can be treated independently given their starting time. The likelihood associated with all journeys in a day has a factorised form

$$p(t_d^{(1)}, \ldots, t_d^{(N)} | s_o^{(1)}, \ldots, s_o^{(N)}) = \prod_{n=1}^{N} p(t_d^{(n)} | s_o^{(n)}) \qquad (2)$$

where $t_d^{(n)}$ is the total travel time of the $n$th user and $N$ the total number of users in a day and each $p(t_d^{(n)} | s_o^{(n)})$ depends only locally on the model parameters, *i.e.* on the delay functions associated with the nodes crossed by the corresponding path. The drawback is that an exact computation of (2) is intractable and one needs approximate inference methods to identify the model parameters from the data.

We address the inference problem in two complementary ways. The first one is a model-approximation method, where we perform the exact inference of the approximate (tractable) model

$$t(\gamma_i) \sim t(\gamma_{i-1}) + \text{Poisson}(a(\gamma_{i-1}, s_o + \bar{t}_{i-1}, \gamma)) \qquad i = 1, \ldots, \ell \qquad (3)$$

where $\bar{t}_{i-1}$ is a deterministic function of the model parameters that is defined by the difference equation

$$\bar{t}_i = \bar{t}_{i-1} + a(\gamma_{i-1}, s_o + \bar{t}_{i-1}, \gamma) \qquad i = 1, \ldots, \ell \qquad (4)$$

The second one is a variational inference approach where we maximise a lower bound of the intractable likelihood associated with (1). In both cases, we use stochastic gradient updates to solve iteratively the corresponding non-convex optimization. Since the closed form solution of (4) is in general not available, the gradients of the objective functions cannot be computed explicitly. At each iteration, they are obtained recursively from a set of difference equations derived from (4), following a scheme that can be seen as a simple version of the back-propagation method used to train neural networks. Finally, we initialize the iterative algorithms by means of a method of moments estimation of the time-independent part of the delay functions. Choosing a random distribution over the feasible paths, this is obtained from the empirical moments of the travel time distribution (of the approximate model (10)) by solving a convex optimization problem.

**London underground experiments** The predictive power of our model is tested via a series of synthetic and real-world experiments based on the London underground network. All details of the multi-layer structure of the network can be found in [36]. In the training step we use input-output data that contain the origin, the destination, the starting time and the exit time of each (pseudonymised) user of the system. This kind of data are produced nowadays by tap-in/tap-out smart card systems such as the Oyster Card systems in London [19]. The trained models can then used to predict the unobserved number of people travelling through a given station at a given time in the day, as well as the user's path preferences for given origin-destination pair. In the synthetic experiments, we compared the model estimations with the values produced by the 'ground-truth' (a set of random parameters used to generate the synthetic data) and test the performance of the two proposed inference methods. In the real-world experiment, we used original pseudonymised data provided by Transport for London. The dataset consisted of more than 500000 origin destination records, from journeys realised in a single day on the busiest part of the London underground network (Zone 1 and 2, see [36]), and a subset of NetMIS records [37] from the same day. NetMIS data contain realtime information about the trains transiting through a given station and, for an handful of major underground stations (all of them on the Victoria line), include quantitative estimation of the realtime train weights. The latter can be interpreted as a proxy of the realtime (unobserved) number of people travelling through the corresponding nodes of the network and used to evaluate the model's predictions in a quantitative way. The model has also been tested on a out-of-sample Oyster-card dataset by comparing expected and observed travel time between a selection of station pairs. Unfortunately, we are not aware of any existing algorithm that could be applicable for a fair comparison on similar settings.

## 2 Travel time model

Let $o$, $d$ and $s_o$ be the origin, the destination and the starting time of a user travelling through the system. Let $\Gamma_{od}$ be the set of all feasible paths between $o$ and $d$. Then the probability of observing a

travel time $t_d$ is a mixture of probability distributions

$$p(t_d) = \sum_{\gamma \in \Gamma_{od}} p_{\text{path}}(\gamma)p(t_d|\gamma) \qquad p_{\text{path}}(\gamma) = \frac{e^{-L(\gamma)}}{\sum_{\gamma \in \Gamma_{od}} e^{-L(\gamma)}} \tag{5}$$

where the conditional $p(t_d|\gamma)$ can be interpreted as the travel time probability over a particular path, $p_{\text{path}}(\gamma)$ is the probability of choosing that particular path and $L(\gamma)$ is some arbitrary 'effective length' of the path $\gamma$. According to (1), the conditional probabilities $p(t_d|\gamma)$ are complicated convolutions of Poisson distributions. An equivalent but more intuitive formulation is

$$t_d = \sum_{i=2}^{\ell(\gamma)} r_i \qquad r_i \sim \text{Poisson}(a(\gamma_{i-1}, s_o + \sum_{k=2}^{\ell} r_k, \gamma)) \qquad \gamma \sim P_{\text{path}}(L(\gamma)) \tag{6}$$

where the travel time $t_d$ is explicitly expressed as the sum of the local delays, $r_i = t(\gamma_i) - t(\gamma_{i-1})$, along a feasible path $\gamma \in \Gamma_{od}$. Since the time at the intermediate positions, *i.e.* $t(\gamma_i)$ for $i \neq 1, \ell$, is not observed, the local delays $r_2, \dots, r_{\ell(\gamma)}$ are treated as hidden variables. Letting $\bar{\ell} = \max_{\gamma \in \Gamma_{od}} \ell(\gamma)$, the complete likelihood is

$$p(r_1, \dots, r_{\bar{\ell}}, \gamma) = p(r_1, \dots, r_{\bar{\ell}}|\gamma)p_{\text{path}}(\gamma) \qquad p(r_1, \dots, r_{\bar{\ell}}|\gamma) = \prod_{i=1}^{\bar{\ell}} \frac{e^{-\lambda_i} \lambda_i^{r_i}}{r_i!} \tag{7}$$

where $\lambda_i = a(\gamma_{i-1}, s_o + \sum_{k=2}^{i-1} r_k, \gamma)$ if $i \leq \ell(\gamma)$ and $\lambda_i = 0$ if $i > \ell(\gamma)$. Marginalizing over all hidden variables one obtains the explicit form of the conditional probability distributions in the mixture (5), *i.e.*

$$p(t_d|\gamma) = \sum_{r_2=0}^{\infty} \cdots \sum_{r_{\bar{\ell}}=0}^{\infty} \delta(t_d - \sum_{i=2}^{\bar{\ell}} r_i) \prod_{i=2}^{\bar{\ell}} \frac{e^{-\lambda_i} \lambda_i^{r_i}}{r_i!} \tag{8}$$

Since $\lambda_i = \lambda_i(r_{i-1}, \dots, r_2)$ for each $i = 2, \dots, \ell$, the evaluation of each conditional probabilities requires performing a $(\ell - 1)$-dimensional infinite sum, which is numerically intractable and makes an exact maximum likelihood approach infeasible. [2]

## 3 Inference

An exact maximum likelihood estimation of the model parameters in $a(x, s, \gamma)$ and $L(\gamma)$ is infeasible due to the intractability of the evidence (8). One possibility is to use a Monte Carlo approximation of the exact evidence (8) by sampling from the nested Poisson distributions. In this section we propose two alternative methods that do not require sampling from the target distribution. The first method is based on the exact inference of an approximate but tractable model. The latter depends on the same parameters as the original one (the 'reference' model (6)) but is such that the local delays become independent given the path and the starting time. The second approach consists of an approximate variational inference of (6) with the variational posterior distribution defined in terms of the deterministic model (4).

### 3.1 Exact inference of an approximate model

We consider the approximation of the reference model (6) defined by

$$t_d = \sum_{i=2}^{\ell(\gamma)} r_i \qquad r_i \sim \text{Poisson}(a(\gamma_{i-1}, s_o + \bar{t}_{i-1}, \gamma)) \qquad \gamma \sim P_{\text{path}}(L(\gamma)) \tag{10}$$

$$\langle t_d^n \rangle = \sum_{t=0}^{\infty} t^n \, p(t) = \sum_{\gamma \in \Gamma_{od}} p_{\text{path}}(\gamma) \sum_{r_2=0}^{\infty} \cdots \sum_{r_{\bar{\ell}}=0}^{\infty} (\sum_{i=2}^{\bar{\ell}} r_i)^n \prod_{i=2}^{\bar{\ell}} \frac{e^{-\lambda_i} \lambda_i^{r_i}}{r_i!} \tag{9}$$

is also intractable.

where the $\bar{t}_i$ are obtained recursively from (4). In this case, the $\ell(\gamma) - 1$ local delays $r_i$ are decoupled and the complete likelihood is given by

$$p(r_1, \ldots, r_{\bar{\ell}}, \gamma) = p(r_1, \ldots, r_{\bar{\ell}} | \gamma) p_{\text{path}}(\gamma) \qquad p(r_1, \ldots, r_{\bar{\ell}} | \gamma) = \prod_{i=1}^{\bar{\ell}} \frac{e^{-\lambda_i} \lambda_i^{r_i}}{r_i!} \qquad (11)$$

where $\lambda_i = a(\gamma_{i-1}, s_o + \bar{t}_{i-1}(\gamma), \gamma)$ if $i \leq \ell(\gamma)$ and $\lambda_i = 0$ if $i > \ell(\gamma)$. Noting that $t_d$ is the sum of independent Poisson random variables, we have

$$p(t_d) = \sum_{\gamma \in \Gamma_{od}} p_{\text{path}}(\gamma) \sum_{r_2=0}^{t_d} \ldots, \sum_{r_{\bar{\ell}}=0}^{t_d} \delta(t_d - \sum_{i=2}^{\bar{\ell}} r_i) \prod_{i=2}^{\bar{\ell}} \frac{e^{-\lambda_i} \lambda_i^{r_i}}{r_i!} = \sum_{\gamma \in \Gamma_{od}} p_{\text{path}}(\gamma) \frac{e^{-\bar{t}_{\bar{\ell}}} \bar{t}_{\bar{\ell}}^{t_d}}{t_d!} \qquad (12)$$

where we have used $\sum_{i=2}^{\bar{\ell}} \lambda_i = \bar{t}_{\bar{\ell}}$. The parameters in the model function $a$ and $L$ can then be identified with the solution of the following non-convex maximization problem

$$\max_{a,L} \sum_{o=1}^{D} \sum_{d=1}^{D} \sum_{s_o=0}^{T-1} \sum_{s_d=so}^{T} N(o, d, s_o, s_d) \log p(s_d - s_o) \qquad (13)$$

where $N(o, d, s_o, s_d)$ is the number of users travelling from $o$ to $d$ with enter and exit time $s_o$ and $s_d$ respectively.

## 3.2 Variational inference of model the original model

We define the approximate posterior distribution

$$q(r, \gamma) = q(r|\gamma) q_{\text{path}}(\gamma) \qquad q(r|\gamma) = p_{\text{multi}}(r; t_d, \eta) \qquad q(\gamma) = \frac{e^{-\tilde{L}(\gamma, t_d)}}{\sum_{\gamma \in \Gamma_{od}} e^{-\tilde{L}(\gamma, t_d)}} \qquad (14)$$

where we have defined $r = [r_2, \ldots, r_{\bar{\ell}}]$, $\eta_i = \frac{\bar{t}_i - \bar{t}_{i-1}}{\bar{t}_{\bar{\ell}}}$, with $\bar{t}_i = \bar{t}_{i-1}$ for all $\ell(\gamma) < i \leq \bar{\ell}$, $p_{\text{multi}}(r; t_d, \eta) = \delta(t_d - \sum_{i=2}^{\bar{\ell}} r_i) t_d! \prod_{i=2}^{\bar{\ell}} \frac{\eta_i^{r_i}}{r_i!}$ and the function $\tilde{L}(\gamma, t_d)$ depends on the path, $\gamma$, and the observed travel time, $t_d$. Except for the corrected length $\tilde{L}(\gamma, t_d)$, the variational distribution (14) share the same parameters over all data points and can be used directly to evaluate the likelihood lower bound (ELBO) $\mathcal{L} = E_q(\log p(t_d)) - E_q(\log q)$ [3]. One has

$$\mathcal{L}(o, d, s_o, t_d) = -\log t_d! + \sum_{\gamma \in \Gamma_{od}} q_{\text{path}}(\gamma) \log \frac{p_{\text{path}}(\gamma)}{q_{\text{path}}(\gamma)} + \sum_{\gamma \in \Gamma_{od}} q_{\text{path}}(\gamma) \sum_{i=2}^{\bar{\ell}} \mathcal{L}_i(\gamma)$$

$$\mathcal{L}_i(\gamma) = \sum_{r_2=1}^{t_d} \ldots, \sum_{r_{\bar{\ell}}=1}^{t_d} p_{\text{multi}}(r; t_d, \eta) \sum_{i=2}^{\bar{\ell}} (-\lambda_i + r_i \log \frac{\lambda_i}{\eta_i}) \qquad (15)$$

with $\lambda_i = a(\gamma_{i-1}, s_o + \sum_{k=2}^{i-1} r_k)$ and $\eta_i = \frac{a(\gamma_{i-1}, s_o + \bar{t}_{i-1})}{\bar{t}_{\bar{\ell}}}$ if $i \leq \ell(\gamma)$ and $\lambda_i = 0 = \eta_i$ if $i > \ell(\gamma)$. The exact evaluation of each $\mathcal{L}_i(\gamma)$ is still intractable due to the multidimensional sum. However, since for any $\gamma$ and $i = 2, \ldots, \ell$, $\lambda_i$ depends only on the 'previous' delays and we can define

$$\eta_{\text{past}} = \frac{\bar{t}_{i-1}}{\bar{t}_{\bar{\ell}}} \qquad \eta_{\text{future}} = \frac{\bar{t}_{\bar{\ell}} - \bar{t}_i}{\bar{t}_{\bar{\ell}}} \qquad \lambda_i = a(\gamma_{i-1}, s_o + r_{\text{past}}) \qquad (16)$$

where $r_{\text{past}} = r_2 + \cdots + r_{i-1}$ and $r_{\text{future}} = r_{i+1} + \cdots + r_{\bar{\ell}}$, and by the grouping property of the multinomial distribution we obtain

$$\mathcal{L}_i(\gamma) = \sum_{r_{\text{future}}=1}^{t_d} \sum_{r_i=1}^{t_d} p_{\text{multi}}(r^{(i)}, t_d, \eta^{(i)}) (-\lambda_i + r_i \log \frac{\lambda_i}{\eta_i}) \qquad (17)$$

where $r^{(i)} = [r_{\text{past}}, r_i, r_{\text{future}}]$ and $\eta^{(i)} = [\eta_{\text{future}}, \eta_i, \eta_{\text{past}}]$. Every $\mathcal{L}_i(\gamma)$ can now be computed in $O(t_d^3)$ operations and the model parameter identified with the solution of the following non-convex optimization problem

$$\max_{a,L,\tilde{L}} \sum_{o=1}^{D} \sum_{d=1}^{D} \sum_{s_o=0}^{T-1} \sum_{s_d=so}^{T} N(o, d, s_o, s_d) \mathcal{L}(o, d, s_o, s_d - s_o) \qquad (18)$$

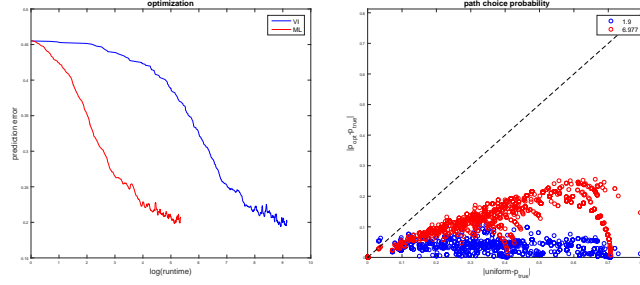

Figure 1: On the left, stochastic iterative solution of (18) (VI) and (13) (ML) for the synthetic dataset. At each iteration, the prediction error is obtained on a small out-of-sample dataset. On the right, distance from the ground-truth of the uniform distribution (x-axis) and the models' path probability (y-axis) for various origin-destination pairs. In the legend box, total distance from the ground-truth.

**Stochastic gradient descent**  Both (13) and (18) consist of $O(D^2T^2)$ terms and the estimation of the exact gradient at each iteration can be expansive for large networks $D >> 1$ or fine time resolutions $T >> 1$. A common practice in this case is to use a stochastic approximation of the gradient where only a random selection of origin-destination pairs and starting times are used. Note that each $\mathcal{L}(o, d, s_o, t_d)$ depends on $a(x, s, \gamma)$ only if the location $x$ is crossed by at least one of the feasible paths between $o$ and $d$.

**Initialization**  The analytic form of the first moments of (12), $\langle t_d \rangle_{s_o} = \sum_{t_d=1}^{\infty} t_d p(t_d) = \sum_{\gamma \in \Gamma_{od}} p_{\text{path}}(\gamma) \bar{t}_{\ell(\gamma)}$, can be used to obtain a partial initialization of the iterative algorithms via a simple moment-matching technique. We assume that, averaging over all possible starting time, the system behaves like a simple communication network with constant delays at each nodes or, equivalently, that $a(x, s, \gamma) = \alpha(x) + V(x, s, \gamma)$, with $\sum_{s=0}^{T} V(x, s, \gamma) = 0$. In this case an initialization of $\alpha(x)$ is obtained by solving

$$\min_{\alpha} \sum_{o=1}^{D} \sum_{d=1}^{D} (t_{od} - \sum_{\gamma \in \Gamma_{od}} p_{\text{path}}(\gamma) \sum_{k=1}^{\ell(\gamma)-1} \alpha(\gamma_k))^2 \qquad (19)$$

where $t_{od} = \frac{1}{Z} \sum_{s_o=0}^{T-1} \sum_{s_d=s_o}^{T} N(o, d, s_o, s_d)(s_d - s_o)$, with $Z = \sum_{s_o=0}^{T-1} \sum_{s_d=s_o}^{T} N(o, d, s_o, s_d)$, is the 'averaged' empirical moments computed from the data. Note that (19) is convex for any fixed choice of $p_{\text{path}}(\gamma)$.

**Total derivatives**  All terms in (13) and (18) are in the form $g = g(\xi, \bar{t}_i)$, where $\xi$ denotes the model parameters and $\bar{t}_i = \bar{t}_i(\xi)$ is defined by the difference equation (4). Since $\bar{t}_i$ is not available as an explicit function of $\xi$ it is not possible to write $g = g(\xi)$ or compute directly its gradient $\nabla_\xi g$. A way out is to compute the total derivative of the function $g$ with respect to $\xi$, *i.e.*

$$\frac{dg(\xi, \bar{t}_i)}{d\xi} = \frac{\partial g(\xi, \bar{t}_i)}{\partial \xi} + \frac{\partial g(\xi, \bar{t}_i)}{\partial \bar{t}_i} \frac{d\bar{t}_i}{d\xi} \qquad (20)$$

where $\frac{d\bar{t}_i}{d\xi}$, for $i = 1, \ldots, \ell$, can be obtained from the iterative integration of

$$\frac{d\bar{t}_i}{d\xi} = \frac{d\bar{t}_{i-1}}{d\xi} + \frac{\partial a(x, s, \gamma))}{\partial \xi} + \frac{\partial a(x, s, \gamma))}{\partial s}\bigg|_{s=\bar{t}_{i-1}} \frac{d\bar{t}_{i-1}}{d\xi} \qquad i = 1, \ldots, \ell \qquad (21)$$

which is implied by (4).

## 4  Experiments

The method described in the previous sections is completely general and, except for the initialization step, no special form of the model functions is assumed. In order to captures few key features of

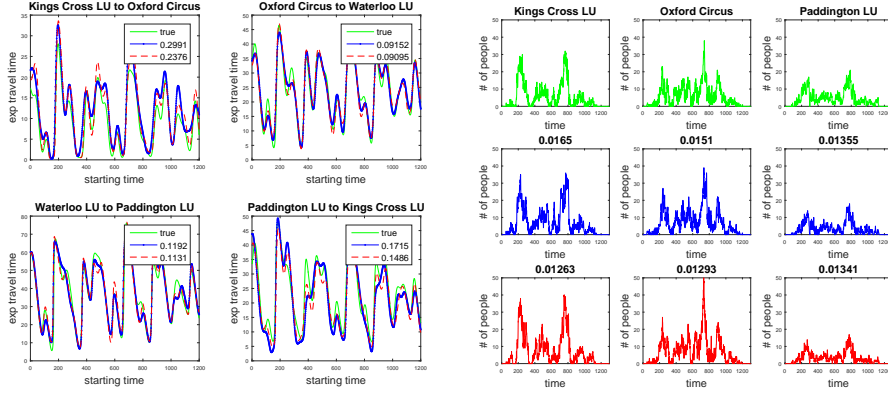

Figure 2: On the left, travel time predicted by the VI model (in blue) and the ML model (in red) of Figure 1 and the ground-truth model (in green) plotted against the starting time for a selection of origin-destination pairs. In the legend box, normalised total distance ($\|v_{\exp} - v_{\mathrm{true}}\|/\|v_{\mathrm{true}}\|$) between model's and ground-truth's predictions. On the right, station loads predicted by the ground-truth (in green) and the VI model (in blue) and ML model (in red) of Figure 1. The three models and a reduced dataset of $N = 10000$ true origin, destination and starting time records has been used to simulate the trajectories of $N$ synthetic users. For each model, the $N$ simulated trajectories give the users expected positions at all times (the position is set to 0 if the users is not yet into the system or has finished his journey) that have been used to compute the total number of people being at a given station at a given time. The reported score is the total distance between model's and ground-truth's normalised predictions. For station $x$, the normalised load vector is $v_x/1^T v_x$ where $v_x(s)$ is the number of people being at station $x$ at time $s$.

a large transportation system and apply the model to the tomography of the London underground, we have chosen the specific parametrization of the function $L(\gamma)$ and $a(x, s, \gamma)$ given in Section 4.1. The parametrised model has then been trained and tested on a series of synthetic and real-world datasets as described in Section 4.2.

## 4.1 Parametrization

For each origin $o$ and the destination $d$, we have reduced the set of all feasible paths, $\Gamma_{od}$, to a small set including the shortest path and few perturbations of the shortest path (by forcing different choices at the line-change points). Let $\mathcal{C}(\gamma) \in \{0, 1\}^\ell$ such that $\mathcal{C}(\gamma_i) = 1$ if the user changes line at $\gamma_i$ and zero otherwise. To parametrize the path probability (5) we chose $L(\gamma) = \beta_1 \ell(\gamma) + \beta_2 c(\gamma)$ where $\ell(\gamma) = |\gamma|$, $c(\gamma) = \sum_i \mathcal{C}(\gamma_i)$ and $\beta_1, \beta_2 \in \mathbf{R}$ are free parameters. The posterior-corrected effective length $\tilde{L}(\gamma, t_d)$ in (14) was defined as

$$\tilde{L}(\gamma) = \tilde{\beta}_\ell \ell(\gamma) + \tilde{\beta}_c c(\gamma) \qquad \tilde{\beta}_i = \theta_{i1} + \theta_{i2} u + \theta_{i3} u^{-1} \qquad u = \hat{t}_d^{-2}(t_d - \hat{t}_d)^2 \qquad i = \ell, c \quad (22)$$

where $t_d$ is the observed travel time, $\hat{t}_d = \sum_{o,d,s_o,s_d} N(o, d, s_o, s_d)(s_d - s_o)$ and $\theta_{ij} \in \mathbf{R}$, $i = \ell, c$ and $j = 1, 2, 3$, are extra free parameters. A regularization term $\lambda(\|\beta\|^2 + \sum_{i=\ell,c} \|\theta_i\|^2)$, with $\lambda = 1/80$, has been added to help the convergence of the stochastic algorithm. We let the local time-dependent delay at location $x$ and time $s$ be $a(x, s, \gamma) = \mathrm{softplus}(\alpha(x) + V(x, s) + W(x, \gamma))$ with

$$V = \sum_{i=1}^{N_\omega} \sum_{j=1}^{N_\phi} \sigma_{ij}(x) \cos(\omega_i s + \phi_j) \qquad W = \sum_{i=1}^{\ell} \rho(x)\delta_{x,\gamma_i}\mathcal{C}(\gamma_i) + \eta(x)\left(\delta_{x,\gamma_1} + \delta_{x,\gamma_\ell}\right) \quad (23)$$

where $\alpha(x), \rho(x), \eta(x) \in \mathbf{R}$ and $\sigma(x) \in \mathbf{R}^{N_\omega \times N_\phi}$ are free parameters and $\{\omega_1, \ldots \omega_{N_\omega}\}$ and $\{\phi_1, \ldots \phi_{N_\phi}\}$ two sets of library frequencies and phases. In the synthetic simulation, we have restricted the London underground network [36] to Zone 1 (63 stations), chosen $N_\omega = 5 = N_\psi$ and set $W = 0$. For the real-data experiments we have considered Zone 1 and 2 (131 stations), $N_\omega = 10$, $N_\psi = 5$ and $W \neq 0$.

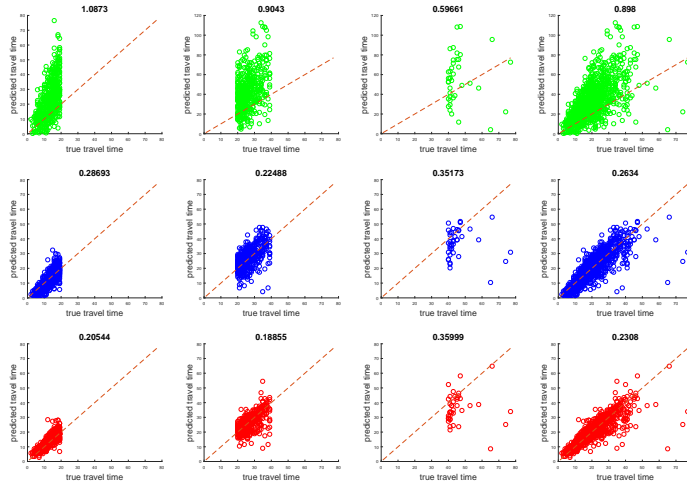

Figure 3: Travel times predicted by a random model (top), the initialization model (middle) obtained from (19) and the ML model (bottom) are scattered against the observed travel times of an out-of-sample test dataset (real data). The plots in the first three columns show the prediction-error of each model on three subsets of the test sample, $\mathcal{S}_{\text{short}}$ (first column), $\mathcal{S}_{\text{medium}}$ (second column) and $\mathcal{S}_{\text{long}}$ (third column), consisting respectively of short, medium-length and long journeys. The plots in the last column show the prediction error of each model on the whole test dataset $\mathcal{S}_{\text{all}} = \mathcal{S}_{\text{short}} + \mathcal{S}_{\text{medium}} + \mathcal{S}_{\text{long}}$ The reported score is the relative prediction error for the corresponding model and subset of journeys defined as $\|v_{\text{exp}} - v_{\text{true}}\|/\|v_{\text{true}}\|$, with $v_{\text{exp}}(n)$ and $v_{\text{true}}(n)$ being the expected and observed travel times for the $n$th journey in $\mathcal{S}_i$, $i \in \{\text{short}, \text{medium}, \text{long}, \text{all}\}$.

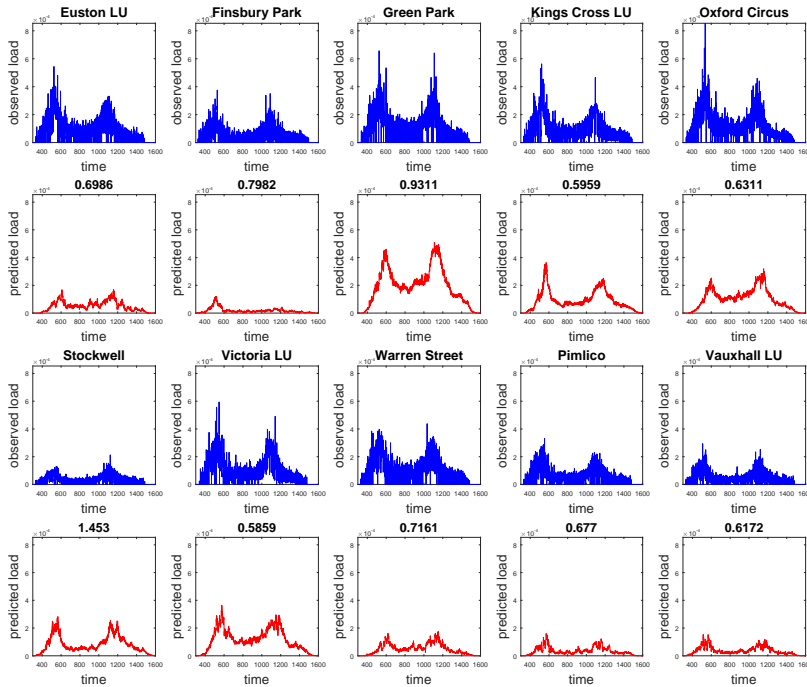

Figure 4: Station loads obtained from NetMIS data (in blue) and predicted by the model (in red). NetMIS data contain information about the time period during which a train was at the station and an approximate weight-score of the train. At time $s$, a proxy of the load at a given station is obtained by summing the score of all trains present at that station at time $s$. To make the weight scores and the model predictions comparable we have divided both quantities by the area under the corresponding plots (proportional to the number of people travelling through the selected stations during the day). The reported score is the relative prediction error $\|v_{\text{exp}} - v_{\text{true}}\|/\|v_{\text{true}}\|$, with $v_{\text{exp}}(s)$ being the (normalised) expected number of people being at the station at time $s$ and $v_{\text{true}}(s)$ the (normalised) weight-score obtained from the NetMIS data.

## 4.2 Methods and discussion

Synthetic and real-world numerical experiments have been performed to: (i) understand how reliable is the proposed approximation method compared to more standard approach (variational inference), (ii) provide quantitative tests of our inference algorithm on the prediction of key unobservable quantities from a ground-truth model and (iii) assess the scalability and applicability of our method by modelling the traffic of a large-scale real-world system. Both synthetic and real-world experiments were are based on the London underground network [36]. Synthetic data were generated from the true origins, destinations and starting times by simulating the trajectories with the ground-truth (random) model described in Section 4.1. On such dataset, we have compared the training performance of the variational inference and the maximum likelihood approaches by measuring the prediction error on an out-of-sample dataset at each stochastic iteration (Figure 1, right). The two trained models have then been tested against the ground-truth on predicting (i) the total travel time (Figure 2, left), (ii) the shape of the users' path preferences (Figure 1, right) and (iii) the local loads (Figure 2, right). In the real-world experiments, we have trained the model on a dataset of smart-card origin-destination data (pseudonymised Oyster Card records from 21st October 2013 provided by Transport for London[4] ) and then tested the prediction of the total travel time on a small out-of-sample set of journeys (Figure 3) . In this case we have compared the model prediction with its indirect estimation obtained from NetMIS records of the same day (Figure 4). NetMIS data contain a partial reconstruction of the actual position and weights of the trains and it is possible to combine them to estimate the load of a given station an any given time in the day. Since full train information was recorded only on one of the 11 underground lines of the network (the Victoria Line), we have restricted the comparison to a small set of stations.

The two inference methods (VI for (18) and ML for (13)) have obtained good and statistically similar scores on recovering the ground-truth model predictions (Figure 2). ML has been trained orders of magnitude faster and was almost as accurate as VI on reproducing the users' path preferences (see Figure 1). Since the performance of ML and VI have shown to be statistically equivalent. Only ML has been used in the real-data experiments. On the prediction of out-of-sample travel times, ML outperformed both a random model and the constant model used for the initialization ($a(x, s, \gamma) = \alpha(x)$ with $\alpha(x)$ obtained from (19) with uniform $p_{\text{path}}$). In particular, when all journeys in the test dataset are considered, ML outperforms the baseline method with a 24% improvement. The only sub-case where ML does worse ( 8% less accurate) is on the small subset of long journeys (see Figure 3). These are journeys where i) something unusual happens to the user or ii) the user visits lot of stations. In the latter case, a constant-delay model (as our initialization model) may perform well because we can expect some averaging process between the time variability of all visited stations. Figure 4 shows that ML was able to reproduce the shape and relative magnitude of the 'true' time distributions obtained from the NetMIS data. For a more quantitative comparison, we have computed the normalised distance (reported on the top of the red plots in Figure 4) between observed and predicted loads over the day.

## 5 Conclusions

We have proposed a new scalable method for the tomography of large-scale networks from input output data. Based on the prediction of the users' travel time, the model allows an estimation of the unobserved path preferences and station loads. Since the original model is intractable, we have proposed and compared two different approximate inference schemes. The model hes been tested on both synthetic and real data from the London underground. On synthetic data, we have trained two distinct models with the proposed approximate inference techniques and compare their performance against the ground-truth. Both of them could successfully reproduce the outputs of the ground-truth on observable and unobservable quantities. Trained on real data via stochastic gradient descent, the model outperforms a simple constant-delay model on predicting out-of-sample travel times and produces reasonable estimation of the unobserved station loads. In general, the training step could be made more efficient by a careful design of the mini-batches used in the stochastic optimization. More precisely, since each term in (13) or (18) involves only a very restricted set of parameters (depending on the set feasible paths between the corresponding origin and destination), the inference could be radically improved by stratified sampling techniques as described for example in [40, 41, 42].

**Acknowledgments**

We thank Transport for London for kindly providing access to data. This work has been funded by a EPSRC grant EP/N020723/1. RS also acknowledges support by The Alan Turing Institute under the EPSRC grant EP/N510129/1 and the Alan Turing Institute-Lloyd's Register Foundation programme on Data-Centric Engineering.

## Footnotes

[1]Other options include negative binomial and shifted geometric distributions

[2] An exact evaluation of the moments

[3]Similar 'amortised' approaches have been used elsewhere to make the approximate inference scalable [38, 39]

[4] The data shown in Figure 3 and 4 are not publicly available, but a reduced database containing similar records can be downloaded from [19]

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
