[Reviews · NeurIPS 2017]

Reviewer 1



The authors develop a novel model for a real-world problem of predicting the number of people traveling through stations at a given time, given dataset of origin-destination pairs of timestamps. The paper presents maximum-likelihood inference for an approximate model (to make it tractable) and variational inference for the true model. The ML method is shown to perform adequately on synthetic data. On real data visualizations (fig 3 and 4) of predictions on a massive real dataset are provided as evidence that it works well. The model is bound to be of much interest to network researchers. The main criticism of the paper is it's evaluation of the predictive ability of their method on the real dataset is lacking. They present scatter plots and time-series requiring the user to verify visually that their method works. These need to be quantified (using e.g., predictive L1 error or something more appropriate), even if only a proxy for the ground-truth is available. Competing methods (e.g., [18]) are not evaluated on the real data; if they're not appropriate for this data, it needs to be addressed in the text. They need to discuss when their method works and when it fails using exploratory analysis. Further, they present the VI method even though they appear not to have succeeded in scaling it to the size of the real dataset. 1. what are the units for figure 2? minutes? is the x-axis "starting time"? 2. you mention stochastic gradient descent in 173-178 but it's unclear if this was actually implemented or used in evaluation? 3. in figure 3, the scatter plots suggest that the initialization does as well as the ML method? what is going on here? 4. unnormalized distance scores are presented in several figures. these are hard to interpret.

Reviewer 2



I thank the authors for the clarification in their rebuttal. It is even more clear that the authors should better contrast their work with aggregate approaches such as Dan Sheldon's collective graphical models (e.g., Sheldon and Dietterich (2011), Kumar et al. 2013, Bernstein and Sheldon 2016). Part of the confusion came from some of the modeling choices: In equation (1) the travel times added by one station is Poisson distributed?! Poisson is often used for link loads (how many people there are in a given station), not to model time. Is the quantization of time too coarse for a continuous-time model? Treating time as a Poisson r.v. requires some justification. Wouldn't a phase-type distribution(e.g., Erlang) be a better choice for time? Such modeling choices must be explained. The main problem, I fear, is not in the technical details. The paper is missing a bigger insight into inferring traffic loads from OD pairs. Something beyond solving this particular problem as a mixture of Poissons with path constraints. It is important to emphasize that the evaluation is not predictive (of the future) but rather it gives estimates of hidden variables. It is a transductive method as it cannot predict anything beyond the time horizon given in the training data. I am OK with that. Figure 4 is missing a simple baseline using the shortest paths. The results of the model in Figure 5 don't seem to match the data in the smaller stations (e.g., Finchley Road). Minor comments: What is the difference between the distributions Poisson and poisson in eqs (7) and (9)? Figure 3: botton => bottom A. Kumar, D. Sheldon, and B. Srivastava. 2013. Collective Diffusion over Networks: Models and Inference. In Proceedings of International Conference on Uncertainty in Artificial Intelligence. 351–360. Garrett Bernstein, Daniel Sheldon,Consistently Estimating Markov Chains with Noisy Aggregate Data, AISTATS, 2016. Du, J., Kumar, A., and Varakantham, P. (2014, May). On understanding diffusion dynamics of patrons at a theme park. In Proceedings of the 2014 international conference on Autonomous agents and multi-agent systems (pp. 1501-1502). International Foundation for Autonomous Agents and Multiagent Systems.

Reviewer 3



This paper is about using probabilistic modeling to infer traffic patterns along a network given only the origin-destination observations. The motivating example is finding the paths of travelers on the London Underground given their “tap in” and “tap out” stations. The problem studied is of important theoretical interest for graph analysis and has applications to human mobility, computer networks, and more. The model the authors specified was innovative and practical, I particularly liked the conditioning of each leg of the journey on the absolute time (Eq 1) and the initialization procedure suggested. The model was described at just the right level of generality for solving a variety of problems. Here are ideas for improvements. Some of the details of the variational inference procedure (parts of Section 3) could have been moved to an appendix to make way for more room in the empirical evaluation. The empirical evaluation could have been more convincing: I did not see any strong differences between VI and maximum likelihood in the inferred station loads, and there was missing any reasonable benchmark methods. Some of the plots were a little confusing (not to mention small; if they are to be included in the main paper itself, and I think they should, then they need to be readable). For example, what do the columns of Figure 4 represent? Please ensure to label all axes and columns in the plots. It would also be helpful to state the questions that you want the empirical evaluation to answer (other than just “testing the model”).